# Mitochondrial Control Region Variants Related to Breast Cancer

**DOI:** 10.3390/genes13111962

**Published:** 2022-10-27

**Authors:** Jorge Hermilo Vega Avalos, Luis Enrique Hernández, Laura Yareni Zuñiga, María Guadalupe Sánchez-Parada, Ana Elizabeth González Santiago, Luis Miguel Román Pintos, Rolando Castañeda Arellano, Luis Daniel Hernández-Ortega, Arieh Roldán Mercado-Sesma, Felipe de Jesús Orozco-Luna, Raúl C. Baptista-Rosas

**Affiliations:** 1Programa de Licenciatura de Médico Cirujano y Patero, Centro Universitario de Tonalá, Universidad de Guadalajara, Tonalá 44340, Mexico; 2Departamento de Ciencias de la Salud-Enfermedad como Proceso Individual, Centro Universitario de Tonalá, Universidad de Guadalajara, Tonalá 45425, Mexico; 3Departamento de Ciencias Biomédicas, Centro Universitario de Tonalá, Universidad de Guadalajara, Tonalá 45425, Mexico; 4Centro de Investigación Multidisciplinaria en Salud, Centro Universitario de Tonalá, Universidad de Guadalajara, Tonalá 45425, Mexico; 5Centro de Análisis de Datos y Supercómputo, Universidad de Guadalajara, Zapopan 45100, Mexico; 6Hospital General de Occidente, Secretaría de Salud Jalisco, Zapopan 45170, Mexico

**Keywords:** breast cancer, polymorphism, mitochondrial genomic, D310

## Abstract

Breast cancer has an important incidence in the worldwide female population. Although alterations in the mitochondrial genome probably play an important role in carcinogenesis, the actual evidence is ambiguous and inconclusive. Our purpose was to explore differences in mitochondrial sequences of cases with breast cancer compared with control samples from different origins. We identified 124 mtDNA sequences associated with breast cancer cases, of which 86 were complete and 38 were partial sequences. Of these 86 complete sequences, 52 belonged to patients with a confirmed diagnosis of breast cancer, and 34 sequences were obtained from healthy mammary tissue of the same patients used as controls. From the mtDNA analysis, two polymorphisms with significant statistical differences were found: m.310del (rs869289246) in 34.6% (27/78) of breast cancer cases and 61.7% (21/34) in the controls; and m.315dup (rs369786048) in 60.2% (47/78) of breast cancer cases and 38.2% (13/34) in the controls. In addition, the variant m.16519T>C (rs3937033) was found in 59% of the control sequences and 52% of the breast cancer sequences with a significant statistical difference. Polymorphic changes are evolutionarily related to the haplogroup H of Indo-European and Euro-Asiatic origins; however, they were found in all non-European breast cancers.

## 1. Introduction

Breast cancer has ranked first among malignant diseases for more than a decade, and it is one of the leading causes of death in the female population, mainly impacting age groups between the fourth and sixth decade of life and affecting all socioeconomic levels [1,2,3].

In Latin America, there are important regional differences regarding the incidence and prevalence, with a higher frequency of breast cancer in areas with the highest socioeconomic incomes compared with indigenous areas, where the socioeconomic level is lower [4,5]. This evidence is compatible with the epidemiology of breast cancer worldwide, where it affects developed countries with greater incidence and has lower incidence and prevalence rates in third world populations.

In addition to socioeconomic and environmental components, several authors have mentioned its relationship with genetic variants in human populations, with special attention paid to the different mitochondrial haplogroups. However, there is a lack of evidence of mitochondrial haplogroups with sporadic [6] or familial hereditary breast cancer [7].

Many authors have suggested that alterations in mitochondrial DNA (mtDNA) play an important role in carcinogenesis [8,9,10,11,12,13,14,15,16] involving the control region, due to the fact that it contains sequences essential for transcription and replication. Previous studies have suggested that polymorphisms in these noncoding sequences of the control region could play an important role in breast cancer pathogenesis [17,18,19,20,21]. However, some studies suggest that different mitochondrial polymorphisms vary from one population to another, and the available evidence suggests that some polymorphisms are more related to the presence of breast cancer than others [22,23,24,25].

Previous works have focused on variants associated with coding regions related to mitochondrial metabolism. Among the most cited in literature is the polymorphism m.10398A>G, related to the synthesis of the protein NADH-ubiquinone oxidoreductase 3 (ND3), which has been identified as a biomarker in different populations such as in Polish [26], Indian [22,27], Chinese [24,28], and, mainly, Afro-descendant groups [22,29,30,31]. This polymorphism has also been associated with metabolic syndrome and mental disorders in populations of Asiatic origin [32]. However, in a meta-analysis, it was determined that there was no association when analyzing this polymorphism individually, without correlating it with other mitochondrial polymorphisms in women affected by this malignant disease [33]. Likewise, the frequency of two novel polymorphisms in the *D-loop* of the control region, m.16290insT and m.16293delA, was statistically more prevalent in breast cancer patients than in control subjects (position 16290: OR = 6.011, 95% CI = 1.2482–28.8411, *p* = 0.002; position 16293: OR = 5.6028, 95% CI = 1.4357–21.8925, *p* = 0.010). In addition, previous observations found one novel mutation in the ND3-coding region at position 10316 (A > G) in 69% of breast cancer patients but not in the control subjects. The study suggests that two novel polymorphisms in the *D-loop* may be candidate biomarkers for breast cancer diagnosis in Bangladeshi women [23].

Although, in Caucasian American groups of European descent, the variant m.10398A>G is related with an increased risk of breast cancer, other polymorphisms with more statistical significance have been identified such as m.16519T>C located in the control region [33,34,35,36,37,38].

Other studies have focused on m.16519T>C [18,29,30,32,36], particularly mutations in the *D-loop* due to the fact of the increased risk of breast cancer, occurring either singularly or in association with other mitochondrial protein-coding genes alterations such as m.10398A>G, m.13368G>A, or m.14766C>T [32,36]. Moreover, the association of several variants resulted in a significant predictive breast cancer factor. Indeed, m.10398A>G, together with other mutations, such as m.4216T>C, m.9055G>A, m.12308A>G, or m.16519T>C, is considered to increase the risk of breast cancer developing in women [32,37].

Several studies involving associations of specific polymorphisms with cancer risk have been thoroughly analyzed due to the fact that are characterized by erroneous experimental designs, misinterpretations, and low-quality data [39,40]. Previous studies, carried out through the analysis of data published in case-control studies of breast cancer with a phylogenetic-based approach and using complete genome sequences and partial sequences that corresponded mainly to segments from the region of control, found inconsistencies and contradictions in the nonsynonymous polymorphism, m.10398A>G, in breast cancer, mainly in the selection of control groups as well as in the use of inadequate statistics and errors in the nomenclature [6,40,41]. However, due to the fact of its potential usefulness as a diagnostic tool, the study of mitochondrial DNA and its relation to cancer must remain an important focus of oncological biomarker research using adequate study designs, population stratification, and independent replication of the results.

The purpose of the present work was to explore mitochondrial sequences of clinical cases of breast cancer diagnosis by reusing the available information in the public, free-access database, GenBank, and determine the prevalence of polymorphisms associated with this neoplastic disease.

## 2. Materials and Methods

The main objective of the work is completely descriptive and at a very basic level to define differences between variants identified in mtDNA obtained from tissue samples with breast cancer and control samples without evidence of the disease.

The search for complete and partial mtDNA sequences was performed using the National Center for Biotechnology Information’s (NCBI) GenBank, focusing on those stored in the repository Nucleotide database (https://www.ncbi.nlm.nih.gov/nucleotide accessed on 19 September 2021). In searching for complete chromosomes, fragments smaller than 15,400 base pairs were not considered. Sequences of a small size were obtained, considering fragments no greater than 1500 bp.

The search strategy was carried out using the keywords “Homo sapiens”, “mitochondrion”, and “Breast Cancer”, using booleans and filters to select the results and excluding the term “*ancient human remains*” to facilitate the search in the database. As control sequences, using the same approach but adding the keywords “*Eskimo*” and “*Inuit*”, which is considered to be a low-prevalence population of breast cancer. Moreover, we considered that in some identified breast cancer mtDNA sequences, it was possible to identify control individuals used in those previous works and even healthy breast mtDNA sequences obtained from the contralateral breast of the same patients.

Once the sequences were identified in the database, the accompanying metadata were analyzed to validate the diagnosis of breast cancer and eliminate those that did not correspond or did not specify the presence of the neoplastic disease. From the selected sequences, the reference of the work appointment was obtained to evaluate the end date of the sampling and its objectives, which were contained in the experimental design of each particular study.

Using the sequence identification numbers of GenBank, the complete sequences were obtained in FASTA format files. Subsequently, the FASTA files were used as inputs fed into the University of California Santa Cruz Genome Browser (http://genome.ucsc.edu/ accessed on 25 September 2021 [42]) and the sequences were aligned against the revised Cambridge Reference Sequence (rCRS) with the latest available version (GRCh38/hg38 Assembly). The FASTA files were aligned using BLAT, a BLAST-like alignment tool [43], and custom sequence tracks were created in the browser with the mtDNA obtained from breast cancer cases, allowing us to observe the variants in a general way and from a panoramic perspective of the mitochondrial genomic landscape.

The haplotification and accounting of polymorphisms were carried out using the MITOMASTER application (https://www.mitomap.org/foswiki/bin/view/MITOMASTER accessed on 15 October 2021), building a database using a .csv file for tabulation and statistical analysis of frequencies. The criteria for the classification of the different haplogroups can be found in the *Phylotree* database (http://www.phylotree.org/ accessed on 19 October 2021) [44]. Once the haplotype was performed and the variants in the sequences were identified, a database was constructed to quantify the haplogroups, haplotypes, and main subclades in the population analyzed with breast cancer.

To evaluate the presence of phantom polymorphisms and discordance among the different haplogroups, the Haplogrep package was used (available at http://haplogrep.uibk.ac.at/ accessed on 10 December 2021). Furthermore, using it as a reference, the sequence NC_012920, deposited in *GenBank* as an *rCRS* [27], and with the tool Haplogrep 2 (v.2.1.19) accessible in the same platform, we constructed a graphical phylogenetic tree to explore the population structure and identify possible phantom polymorphisms not assigned to the resulting haplogroup. All variants with a score of <3 and which were found in at least two different sequences according to the criteria of Soares et al. [45] were considered.

Only complete mtDNA sequences were analyzed, and a phylogenetic tree was generated, graphically representing the presence of polymorphisms associated to the various haplogroups, global and local private mutations, and regressive mutations as well as the expected loss of polymorphisms for each haplogroup.

For the analysis of the structure of the populations, a single file with all of the sequences in the FASTA format was used as an input for the multiple sequence alignment application with *CLUSTAL W*, using the Molecular Evolutionary Genetics Analysis (MEGA) package (https://www.megasoftware.net/ accessed on 10 January 2022). Once aligned, the tools, including the mtDNA nomenclature, mtSNP profile, and mtSNP frequency, in the package mtDNAprofiler were used (http://mtprofiler.yonsei.ac.kr/ accessed on 10 January 2022). From the use of these tools, changes in the sequences were identified as well as insertion and deletion sites. In addition, all of the sequence alignments were reviewed to detect polymorphic heteroplasmic sites.

The number of total polymorphic loci (mtSNPs), the number of fixed differences, the polymorphic and monomorphic mutations among the populations, the shared polymorphisms, and the average number of different nucleotides between populations were also estimated. In this way, the identification of the most common polymorphisms in the control region and in the coding region of the mtDNA was facilitated.

Specific variants were searched for in the regions associated with MDP mitochondria-derived proteins in the region of the MT-RNR1 gene that transcribes for 12s RNA located at the genomic coordinates chrM:648–1601 (the MOTS-c sequence of 51 bp with localization at chrM:1343–1393) as well as the variants in the MT-RNR2 gene, which transcribes for 16s RNA at the coordinates chrM:1669–3231 (*Humanin* of 75 bp with localization at chrM:2633–2707; *SHLP1* of 75 bp with localization at chrM:2.485–2.559; *SHLP2* with 81 bp with localization at chrM:2088–2168; *SHL3* with 117 bp with localization at chrM:1703–1819; *SHLP4* with 81 bp with localization at chrM:2442–2522; *SHLP5* with 75 bp with localization at chrM:2485–2559; *SHLP6* with 63 bp with localization at chrM:2990–3052).

To evaluate its correlation, the breast cancer group’s frequency was compared with selected polymorphisms in breast cancer mtDNA sequences and controls; the *X*^2^ test was performed using R v.4.2.0 *(*https://cran.r-project.org/ accessed on 8 February 2022) and RStudio v.2022.02.2+485 (https://www.rstudio.com/ accessed on 10 February 2022).

## 3. Results

From the search of the NCBI GenBank database using the previously defined criteria, 124 mtDNA sequences were identified, where 90 complete mtDNA (16.5 kb) sequences were identified, 52 belonging to patients with a confirmed diagnosis of breast cancer, and 34 sequences were used as controls (Appendix A). All the samples used in the analysis were obtained from breast biopsies. These control sequences were distributed in three categories: 9 distant normal tissue sequences, 10 paracancerous normal tissue sequences obtained from Wang et al. [46], and 15 sequences obtained from cells identified as normal under laser capture microdissection from biopsies of patients diagnosed with invasive breast carcinoma reported in the work of Fendt et al. [47]. Another five complete sequences were obtained from Hernandez de la Cruz et al. (2018) and two sequences from Imanishi et al. (2011) by direct submission to the NCBI GenBank (Appendix A).

There were 34 partial sequences (i.e., smaller than 0.5 kb) of breast cancer cases, of which 12 sequences covered the HVR1 region, and 24 sequences were from the HVR2 region of the mitochondrial chromosome. It is important to note that 48.3% (i.e., 60 sequences, where 45 were obtained from breast cancer tissue and 15 were controls) of the sequences were reported in Europe by Gasparre et al. [48] and Fendt et al. [47] (Appendix A). Only 23.3% (i.e., 29 sequences, where 10 were obtained from breast cancer tissue, and 19 were controls) were of Chinese origin, reported by Wang et al. [46], which explains why the haplotyping of the sequences was related with Caucasians via haplogroup H (Appendix A).

Regarding the partial sequences, 14 sequences from the HVR1 region were published by Ghatak et al. [49], and the 24 remaining sequences (12 were from the HVR1 region, and the other 12 were from the HVR2 region) were associated with direct referrals without related publications and provided by Darvishi et al. [50] (Appendix A).

The number of total polymorphic loci, without considering potential phantom variants, was analyzed, and near half of the variations were in the control region with an average of 10.7. The results of haplotyping in the population with breast cancer showed different clades and subclades distributed in 15 haplogroups (Appendix A). Seven possible phantom polymorphisms were identified in the forty-seven complete mtDNA sequences, and they were excluded from the subsequent analyses [51] (Appendix A).

After haplotyping and navigation in the UCSC genome browser, we found 214 different polymorphisms that shared sequences, which we used as controls, and 241 unique polymorphisms were found only in breast cancer, while only 1 polymorphism, A3480G, was found exclusively in a control sample (Appendix A Appendix A).

The cumulative analysis showed that most of the variations in the breast cancer sequences were unique polymorphisms (Figure 1). Only 13 mutations were related in almost 50% of the sequences studied. In contrast, in the control sequences, there were up to 22 different polymorphisms. Most of the variations were found in unique polymorphisms distributed along the mitochondrial genome (Appendix A Appendix A); however, the most frequent repetition was concentrated in the control region, between positions 576 and 16024 of the mtDNA (Appendix A Appendix A).

The most frequently shared polymorphisms were m.310del (rs869289246), found in 61.7% (21/34), and m.16519T>C (rs3937033), found in 58.8% (20/34) of the controls, while m.315dup (rs369786048) was found in 60.2% (47/78) of the breast cancer cases, with a statistically significant difference (*p* < 0.05) (Table 1).

Regarding the variants found in MPD, we identified 11 polymorphisms of which 5 were not reported in the NCBI dbSNP database, but none of them were significantly different between the group with breast cancer and the control sequences (Table 2).

## 4. Discussion

In humans, the mutation rate of the mitochondrial genome is at least ten times higher than that of the nuclear genome. Most mutations in mtDNA accumulate in the displacement loop, called the *D-loop*, in the noncoding control region. The *D-loop* functions as a promoter for both heavy and light strands of mtDNA and, just as the control region, does not encode any functional peptides.

In our analysis, 32 polymorphisms were present in two sequences and 60 sequences with a single polymorphism were present in each sequence. In m.16147C>T, there were two different polymorphisms: a transition C/T in three sequences and C/A in one.

When we compared the relative frequencies of each sequence, we observed that there was a significant proportion of exclusive polymorphisms in the breast cancer cases that were practically absent in the control samples (Appendix A Appendix A).

The most frequent variants in the breast cancer cases, with a lower incidence in the controls, were m.3010G>A, m.16311T>C, m.16189T>C, and m.16519T>C; however, m.16519T>C, the most important polymorphism, represented at least half of the total sequences shared. It is an important point of focus that m.16189T>C and m.16311T>C were positioned in HVR1 and that m.16519T>C was placed in a noncoding position in the control region. Of note, at position 16,189, the sequences HG825995.1, HG826004.1, and GU592040.1 had a deletion that was not registered in the dbSNP database, in addition to an insertion in the sequence AB626609.1 (variant m.16189delinsCC with the dbSNP ID rs369574569).

From the total 86 mtDNA complete sequences obtained from patients diagnosed with breast cancer, six common polymorphisms were identified in the sequences m.263A>G, m.750A>G, m.1438A>G, m.4769A>G, m.8860A>G, and m.15326A>G, representing more than 98% of the analyzed sequences obtained from malignant tissue. However, they were also found in most of the control samples without statistical differences.

The polymorphism m.10398A>G (rs2853826), usually associated with breast cancer in other studies, was found only in 18.0% (9/50) of the breast cancer mtDNA sequences analyzed as well as in 29.4% (10/34) of the control samples, without statistical significance (Table 1). The m.10398A>G variant was found in the coding region of the MT-ND3 gene that codes for subunit ND3 of complex I (NADH dehydrogenase), although no significant differences were identified between the cases with malignant neoplasia and their respective controls (Table 1). The frequencies of m.315dup (rs369786048) and m.16519T>C were the most recurrent polymorphisms found in our analysis, either in isolation or both in the same sequence, with statistical significance (Table 1 and Appendix A Appendix A).

In terms of heteroplasmic changes in mtDNA sequences, our search showed 22 heteroplasmic changes distributed in only 17 complete sequences (Appendix A). On the other hand, partial sequences presented only seven heteroplasmic changes in only four sequences, all obtained by laser capture microdissection (LCM). These changes were not identified in the rest of the sequences with heteroplasmic somatic transitions described previously by Wang et al. (2007) [46]. The polymorphism m.2275T>C was found in the primary cancerous tissue identified as EF114285.1 but not in other normal tissues. The polymorphism m.8601A>G was related to the primary cancerous sequence EF114276.1 and the paracancerous normal sample EF114277.1, obtained from cells without dysmorphic or malignant changes under macro- and microscopic analysis but with polymorphic changes related to the neoplastic genotype.

Other common findings were polymorphisms in the control region that clearly showed a characteristic pattern related to the presence of one polymorphism and the absence of others (Appendix A Appendix A). We were able to identify that in 47 mitochondrial sequences analyzed, all had alterations in positions 309 to 315, characterized by only three situations: the first was manifested by a deletion (m.310del with the dSNP ID rs869289246); the second was a variation characterized by the insertion of three bases in m.310delinsCCTC (rS46492218); and the third was characterized by the translocation of a thymine for cytosine m.310T>C (rs1556422421). An interesting observation was that the changes in m.310T>C (34.6% of cases; 27 of the 78 sequences) and m.315dup (60.2% of the cases; 47 of the 78 sequences) (Table 2) were observed in almost half of the sequences of non-European origin.

When the variant m.315dup was absent, we identified the presence of the polymorphism m.310del. Even though the most frequent polymorphism in position 310 was m.310del, our analysis found m.310T>C in 4 of the 78 sequences (5.1%) and m.310delinsCCTC in 1 of the 78 sequences (1.2%). After the statistical analysis using the *X*^2^ test, m.310del and m.315dup were significantly different when comparing cases and controls; we also ruled out a potential artifact associated with ghost mutations (Appendix A).

Variations in the repeat sequences between nucleotides 303 and 316–318 were defined by D310, although they were related with oxidative stress and resulted in a compensatory increase in the mtDNA copy number of homopolymeric C repeats in the noncoding mitochondrial control region. These variants in D310 were present in 98% of the analyzed sequences. Although most of these polymorphisms were commonly associated with specific haplogroups, it was revealed that the m.315dup polymorphism was evolutionarily associated with haplogroup H of Caucasian origin and infrequent in Indo-European haplogroups, with a frequency lower than 1.5%. In our analysis, 99% of sequences of non-European origin with breast cancer (Appendix A) were present.

D310, a term that defines these changes, previously described in 2001 by Sanchez-Cespedes et al. [52] is also found in several types of tumors and, apparently, it is the most frequent hotspot of mitochondrial genome somatic mutations in various cancers located in the *D-loop* region. D310 refers to a mononucleotide repeat sequence between positions 303–309 and 316–318 of mtDNA with variable lengths of cytosine repetitions [52,53,54,55]. This length of cytosine bases is highly polymorphic, ranging from seven to nine nucleotides; however, usually seven cytosines are the most common finding. The D310 repeat is a part of conserved sequence block II of the hypervariable region II, located 92 bp away from the heavy-strand replication region, and mitochondrial chromosome replication is initiated when this loci forms a persistent RNA–DNA hybrid that results in the initiation of mtDNA heavy-strand replication with the conserved sequence block I and conserved sequence block III regions [55,56,57]. Previous genomic analysis revealed that only a homopolymeric sequence in conserved sequence block II, ranging from six to twelve nucleotides, showed high variability. Transcriptional analyses revealed that most of the common polymers can support accurate transcriptional initiation [58].

Alterations in the mtDNA replication rate can result from a deletion or insertion in the number of C residues in the poly-C repeat, which may interfere with the binding of mtDNA polymerase γ and other trans-acting elements. It is believed that D310 plays a vital role in preserving the number of mtDNA contents. Because the control region is a susceptible region to oxidative stress, many researchers have hypothesized that the D310 polymorphisms are related to an increased stress status.

Apparently, there was no relationship between the different haplogroups associated with risk of or protection against breast cancer. However, when we analyzed the sequence information, it was possible to observe polymorphisms commonly related to cancer. It is important to note that there is a statistical issue related to the fact that most of the samples were of European origin, as they comprised close to half of the population with breast malignant disease (44.9%) and they were more similar to the sequences used as the *rCRS* reference genome in the database and genomic browsers.

Regarding the presence of phantom polymorphisms, defined as systematic artifacts generated during the sequencing process, and despite the common belief that these artificial changes in the sequence are almost ubiquitous in the results, their frequency and impact on the interpretation of the results can vary dramatically [49]. In mtDNA, they generate an access point pattern quite different from that of natural polymorphisms, which were found in the analyzed sequences [45]; therefore, it is important to identify them before making diagnostic inferences or related hypotheses.

Rejecting false polymorphisms as artifacts, taking the phylogenetic knowledge into account, the procedure incorporated all variants with frequency scores based on 2196 complete mitochondrial genomes as references [26]. All of the remaining variants in a sample were annotated, and if a variant occurred with a frequency of less than three and at least two samples shared this change, then it was listed as a rare polymorphism in the report. The reason for this threshold is that a known phantom variant with score of two is, in fact, also present within the rare polymorphism list. Therefore, this procedure allows for the identification of potential phantom changes in the sequences [26].

Otherwise, of the 241 polymorphisms exclusive to the neoplastic tissue, only 9 were repeated in more than three different sequences in our analysis: m.16183A>C repeated in 11 different sequences; m.16217T>C in 5; and m.16298T>C in 4. The rest of the polymorphisms were repeated three times in 6 sequences, twice in 22 sequences, and only once in 193 sequences. Some of these changes in the mtDNA sequences were related in the biomedical literature to other types of malignant neoplastic diseases [46].

In a previous preliminary analysis with fewer sequences, we identified that the polymorphism, m.315dup, was not associated, in the literature, with breast cancer. However, it had as a reference two samples of medium prevalence, which is very low for neoplastic disease [59]. There was a change in the sequence m.315dup, where cytosine was inserted between positions 311 and 315, compared to the *rCRS*; thus, it was translated into these sequences, and the *rCRS* was larger than many other sequences in the mitochondrial genomes (five or six nucleotides at this position) (Appendix A Appendix A).

The polymorphism m.315dup is considered to be one of the most recent mutations, occurring in the last 60,000 years. This is a type of mutation, now well accepted and detailed in many of the phylogenetic trees, which is related to haplogroup H. Although this haplogroup is related to populations of ethnic origin, it has been found to have a high prevalence in Latin American mestizos and in other populations with different haplogroups [60,61,62,63,64].

The m.315dup and m.310T>C polymorphisms, which were found in the noncoding HVR2 fragment of the *D-loop* in the mtDNA control region, are considered as access points due to the high frequency of changes. Furthermore, the modifications of these are allowed and may have implications for the appropriate transcription and regulation of mitochondrial genome expression. This poly-C hotspot area in D310 is considered very polymorphic and can be different between direct relatives by the maternal line [45,65,66,67], and due to its high prevalence in Caucasian groups, m.315dup is usually not considered during bioinformatic analysis, and it is not used in the construction of phylogenetic trees because it is usually identified. However, there are reports in the literature where the m.315insC polymorphism is associated with various forms of cancer and other chronic degenerative diseases [50,66]. This poly-C tract in the mitochondrial *D-loop*, located commonly between the 303 and 315 nucleotides, was identified as a frequent hotspot mutation region in human neoplasia, including breast cancer [60,61], suggesting that mtDNA instability at this site may be a common characteristic in this malignant disease.

The alteration hallmarks of D310, found in all of the analyzed malignant sequences and identified in only two related options, were m.315dup and m.310del, but in neither case were both in the same sequence, and both polymorphisms are evolutionarily related to haplogroup H of Caucasian, Indo-European, and Euro-Asiatic origins. However, that they were found in all of the non-European origin samples with breast cancer could be a clue for future research to explore, suggesting a potential association with specific population groups where there is a higher incidence of breast cancer. Our analysis of D310 identified more frequent variants, including m.315dup and other variants, such as m.309dup (rs878871521), related to insertions of C in positions 303 to 309 (a change of seven to eight cytosines); m.309del (rs1556422420), related to a change of CTC by only one cytosine, with a deletion of CT (left-shifted) or deletion of TC (right-shifted); and m.310T>C (rs1556422421) and m.310del (rs869289246), with the loss of a nucleotide.

The other important finding in our analysis was the presence of m.16519T>C, found in nearly two-thirds of the control sequences and half of the breast cancer sequences with significant statistical differences. Other studies have focused on this polymorphism, apparently, without relation with other variants previously described in the available literature [21,30,31,32,33,37]. Moreover, the association of several variants resulted in a significant predictive breast cancer factor. Indeed, m.10398A>G, usually associated with breast cancer in other studies [24,27,29,30,31,32,49], was found in only 24% of the mtDNA sequences analyzed without statistical significance, and in association with other mutations, such as m.4216T>C, m.9055G>A, m.12308A>G, or m.16519T>C, it was found to increase the risk of a woman developing breast cancer [36].

m.16519T>C was associated in the *Phylotree* database between the branches of haplogroup L5 and haplogroup L2 with 11 other mutations (i.e., m.16311T>C, m.16189T>C, m.16187C>T, m.15301A>G, m.13506C>T, m.13105A>G, m.10810T>C, m.10688G>A, m.8655C>T, m.825T>A, and m.247G>A) back to its CRS value and usually found in Euro-Asiatic mitogenomes. These ancient polymorphisms arose 140,000 years ago and are considered part of the suggested Mitochondrial Eve genome, supporting the widely accepted theory related to first human migration outside of the African continent. Once again, the high number of mutations suggests there was a significant bottleneck in human evolution at the time, perhaps around 120,000 years ago, which might have lasted for many thousands of years.

Finally, the molecular mechanisms underlying the increased risk of cancer due to the presence of these specific mtDNA polymorphisms are still unclear. The control region is important for the regulation of mitochondrial genome replication and expression. The polymorphisms in this region might affect mtDNA replication and lead to electron transport chain alterations, resulting in compensatory increases in glycolytic ATP production. However, one of the inevitable products of these alterations is an increased release of highly reactive oxygen species, which may lead to mitochondrial abnormalities. These abnormalities invoke a mitochondria-to-nucleus retrograde response and, finally, result in nuclear genome damage, which contributes to initial events related to carcinogenesis [68,69,70]. The regulation of mitochondrial genome replication from the control region might also lead to mtDNA damage [71,72,73] and, with a critical number of mitochondrial genome changes, to cellular apoptosis [74], which finally could induce cancer development.

## 5. Conclusions

In conclusion, the changes in the mtDNA breast cancer were polymorphic and not associated with a few unique mutations. Most of the polymorphisms related to breast cancer were unique to each sequence. The combination of the position of the polymorphism provides some clues regarding mitochondrial alterations in breast cancer.

The polymorphism, m.10398A>G, usually associated with breast cancer in other studies, was found only in one-quarter of the mtDNA sequences analyzed, without statistical significance.

Some polymorphisms placed in the control region of the mitochondrial chromosome were present in the breast cancer sequences, but only m.16519T>C was related to the control sequences with statistical significance. Alterations in the poly-cytosine area, defined as D310, were found in all the analyzed malignant sequences. Only two variants related to changes in D310 were found, m.315dup and m.310del, but in neither case were both in the same sequence. These D310 polymorphisms are evolutionarily related to haplogroup H of Caucasian, Indo-European, and Euro-Asiatic origins. However, both were found in all non-European origins with breast cancer. Mitochondrial variants in D310 related with deletions/insertions and base changes mainly involved purine transitions, which were assumed to have occurred after the action of reactive oxygen species. Many frequent variants accumulated in the D-loop, a specialized sequence in the control region that functions as a promoter for both heavy and light strands of mitochondrial chromosomes. A mononucleotide repeat between nucleotides 303 and 316–318 was identified as a frequent hotspot for mutations in primary tumors and in our research.

Among the main limitations of our research, we identified a low number of sequences analyzed and the lack of representation of other population groups can generate selection biases, because in the database consulted most of the individuals are limited. Further analysis is required with the objective to evaluate more sequences and to calculate the correlation with the risk in the development of malignancy. This approach will provide us with a general perspective on the importance of consolidating the evidence in specialized repositories and that have not been crossed yet. In conclusion, these specific variants associated with breast cancer found in haplogroup H of Indo-European and Euro-Asiatic origins and in all different haplogroups non-European are probably related. It is important to include populations of different origins than Caucasian, with the purpose of showing the genetic differences related to breast cancer, regardless of high and low prevalence. With this focus, our understanding of these malignant diseases will improve through the interpretation of the complete mitogenome.

## Figures and Tables

**Figure 1 genes-13-01962-f001:**
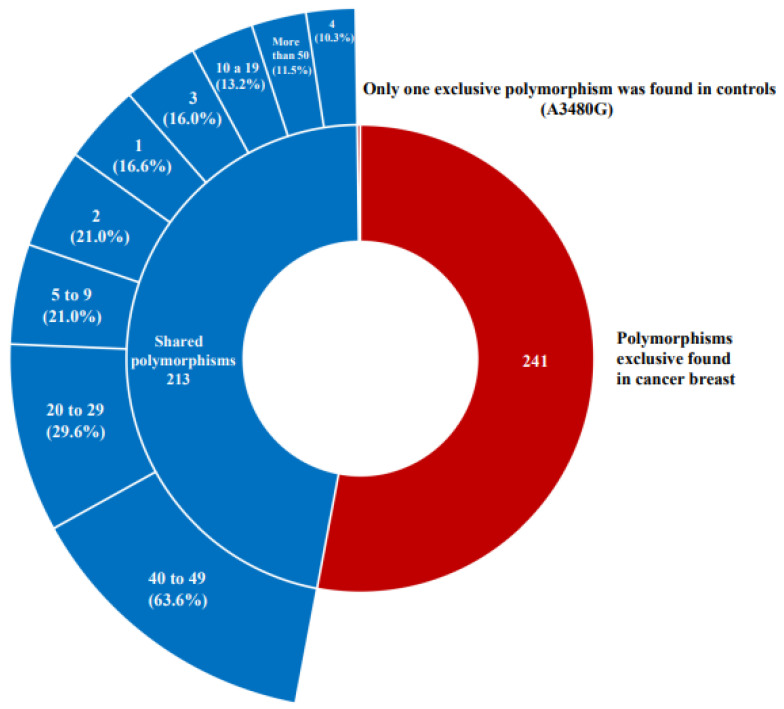
The frequency of the variant repetitions among the different complete sequences obtained from individuals diagnosed with breast cancer and from the control groups.

**Table 1 genes-13-01962-t001:** Frequent polymorphisms in different mtDNA regions of the breast cancer cases and the controls.

mtDNA Region (Positions)	Polymorphism	*NCBI* *dbSNP* ID	Breast Cancer, %	Controls, %	*X* ^2^
	m.310del	rs869289246	34.6 (27/78)	61.7 (21/34)	0.007 *
HRV-II	m.310T>C	rs1556422421	5.1 (4/78)	0 (0/34)	0.178
(57–372)	m.310delinsCCTC	rS46492218	1.2 (1/78)	2.9 (1/34)	0.542
	m.315dup	rs369786048	60.2 (47/78)	38.2 (13/34)	0.030 *
MT-ND3 (10059–10404)	m.10398A>G	rs2853826	18.0 (9/50)	29.4 (10/34)	0.219
HRV-I	m.16183A>C	rs28671493	12.8 (10/78)	14.7 (5/34)	0.787
(16024–16383)	m.16189T>C	rs28693675	18.8 (17/90)	26.5 (9/34)	0.354
	m.16311T>C	rs34799580	12.2 (11/90)	23.5 (8/34)	0.118
	m.16519T>C	rs3937033	51.5 (34/66)	58.8 (20/34)	0.007 *

* The difference was statistically significant between the breast cancer cases and the controls at *p* < 0.05.

**Table 2 genes-13-01962-t002:** Polymorphisms in mitochondrial-derived protein sequences related to breast cancer cases and the controls.

mtDNA Region Genes (Positions)	Polymorphism	*NCBI* *dbSNP* ID	Breast Cancer, %	Controls, %	*X* ^2^
MT-RNR1					
MOTS (1343–1393)	m.1346A>G	rs879104061	1.9 (1/52)	0 (0/34)	0.416
MT-RNR2					
Humanin (2633–2707)	m.2706A>G	rs2854128	61.5 (32/52)	76.4 (26/34)	0.418
SHLP1 (10059–10404)	-	-	-	-	-
SHLP2 (2088–2168)	m.2124A>G	-	1.9 (1/52)	0 (0/34)	0.416
	m.2145del	-	1.9 (1/52)	0 (0/34)	0.416
SHLP3 (1703–1819)	m.1709A>G	rs200251800	3.8 (2/52)	11.8 (4/34)	0.158
	m.1716A>G	rs1556422559	1.9 (1/52)	5.9 (2/34)	0.327
	m.1719A>G	rs3928305	13.4 (7/52)	5.9 (2/34)	0.261
	m.1736A>G	rs193303006	3.8 (2/52)	11.8 (4/34)	0.158
	m.1811A>G	rs28358576	1.9 (1/52)	2.9 (1/34)	0.759
SHLP4 (2442–2522)	m.2445A>G	-	1.9 (1/52)	0 (0/34)	0.416
SHLP5 (2780–2854)	-	-	-	-	-
SHLP6 (2990–3052)	m.3010A>G	rs3928306	21.1 (11/52)	17.6 (6/34)	0.689

## Data Availability

Sequence data are available at https://www.ncbi.nlm.nih.gov/nucleotide/. All the information used in this article is available in public databases at the links provided in Section 2.

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
