# Peer review of "Mitochondrial Control Region Variants Related to Breast Cancer"

_genes, 2022, doi:10.3390/genes13111962_

Round 1
Reviewer 1 Report
The topic addressed in the manuscript is of great interest, since the search for mitochondrial variants associated with breast cancer is of great importance in the discovery of new biomarkers that can be used at the clinical level. Nevertheless, there are many points that are not clear or correctly described in the work, which makes it difficult to consider whether the results obtained and the authors' conclusions are totally correct.
Major Revisions
With regard to the objectives of the work, it is not understood if they want to study mitochondrial variants associated with breast cancer risk or are looking for variants associated with the tumor process (for diagnostic purposes, for example). This issue substantially changes the experimental design.
In my opinion, the authors should explain in the manuscript if the sequences obtained from the patients are from the tumor tissue of patients with breast cancer or from blood (normal tissue). In turn, if the objective is to detect risk markers, the control sequences must come from healthy individuals. In the presented work, many of the control sequences come from normal tissues of individuals with breast cancer or paracancer tissues. Paracancer tissues cannot be considered really normal tissue, because were defined as the tissue less than 2 cm away from the tumor edge.
However, if the objective is to detect markers associated with the tumor process, sequences obtained from healthy tissues of the same patients that were used for the "case" group should be used as control sequences.
That is why in both objectives it would have been recommended to use The Cancer Genome Atlas (TCGA) database, which contains hundreds of sequencing data from patients with breast cancer from blood samples, tumor tissue and normal tissue, and in turn has an extensive database of healthy individuals.
Middle and Minor revisions
1. The Introduction is too long.
2. Results. The number of sequences throughout the Results should be reviewed as there are errors and the figures should be improved:
a. In line 197 “38 sequences were used as control” is incorrect, there are 34.
b. In line 204 and 205: 12 + 24 sequences are not "38".
c. Improve Figure 1 (font size and color references are missing).
d. Supplementary figures must be numbered in the order in which they are cited in the text, and must have legends. Add Y-axis values ​​to charts.
e. Line 261: “From the total 86 mtDNA complete sequences”? Previously, it had been described that there were 52 complete sequences from patients with breast cancer.
f. Line 269, 272 and 274: where it says Table 4, it should say Table 1.
g. Line 295: The data in parentheses correspond to the m.310del variant and not to the m.310T>C variant.
3. In the Results section a lot of discussion is added (after which there is a Discussion and Conclusions section), and in some cases they are described repeatedly.
4. Discussion.
a. The first 7 paragraphs of the Discussion can be shortened quite a bit.
b. The discussion of the difference of this work with the previous work of the authors (Ref. 60) should be expanded, since some sequences are the same.
c. It is not highlighted what this work contributes to what is already reported on mitochondrial variants associated with breast cancer.
d. The fact that the 313.dup and 310del variants were found in all samples of non-European origin is not discussed or hypothesized.
e. The limitations of the study should be described, for example low number of sequences analyzed, etc.
5. Conclusions: they are a summary of the results. The novel contribution of the work should be emphasized.
Author Response
Dear reviewer
We appreciate all your time and effort in reviewing our manuscript and improving it for consideration by the Editorial Committee for publication.
All your comments and recommendations have been addressed as described below:
"The topic addressed in the manuscript is of great interest, since the search for mitochondrial variants associated with breast cancer is of great importance in the discovery of new biomarkers that can be used at the clinical level. Nevertheless, there are many points that are not clear or correctly described in the work, which makes it difficult to consider whether the results obtained and the authors' conclusions are totally correct."
RESPONSE: All points mentioned that are not clear or correctly described in the work were edited and corrected, which facilitates the interpretation of the results obtained and the conclusions.
In abstract section, lines 19-20 where says:
"Our purpose was to explore mitochondrial sequences of cases with breast cancer from different origins and determine the associated polymorphisms."
It was edited as follows:
"Our purpose was to explore differences in mitochondrial sequences of cases with breast cancer and compared with controls samples from different origins."
Major Revisions
"With regard to the objectives of the work, it is not understood if they want to study mitochondrial variants associated with breast cancer risk or are looking for variants associated with the tumor process (for diagnostic purposes, for example). This issue substantially changes the experimental design.
RESPONSE: The experimental design was modified and described in greater detail in the materials and methods section, following the reviewer's recommendations.
The main objective of the work is completely descriptive and at a very basic level to define differences between variants identified in mtDNA obtained from tissue samples with breast cancer and control samples without evidence of the disease."
"In my opinion, the authors should explain in the manuscript if the sequences obtained from the patients are from the tumor tissue of patients with breast cancer or from blood (normal tissue)."
RESPONSE: It was defined that all the samples used in the analysis were obtained from breast biopsies.
In Results section, line 205 it was edited adding this aclaration:
"All the samples used in the analysis were obtained from breast biopsies."
"In turn, if the objective is to detect risk markers, the control sequences must come from healthy individuals. In the presented work, many of the control sequences come from normal tissues of individuals with breast cancer or paracancer tissues. Paracancer tissues cannot be considered really normal tissue, because were defined as the tissue less than 2 cm away from the tumor edge."
RESPONSE: The main objetive ws explore frequent variants, not define biomarkers. It was defined that all the samples used in the analysis were obtained from breast biopsies.
"However, if the objective is to detect markers associated with the tumor process, sequences obtained from healthy tissues of the same patients that were used for the "case" group should be used as control sequences."
That is why in both objectives it would have been recommended to use The Cancer Genome Atlas (TCGA) database, which contains hundreds of sequencing data from patients with breast cancer from blood samples, tumor tissue and normal tissue, and in turn has an extensive database of healthy individuals.
RESPONSE: It was added to the manuscript to the clarification that the initial objective in this work was to define if there was a difference between the mitochondrial variants in tissue obtained from biopsies of malignant breast tumors and control samples obtained from paraneoplastic tissue and healthy breast tissue from the same patients.
We take into account its recommendation for the following work where sequences obtained from different non-diseased controls are compared in a different experimental design.
Middle and Minor revisions
1. The Introduction is too long. RESPONSE: The paper introduction was edited to shorter version: Original introduction of 987 words (line 34 to 116) reduces to 775 words in the final draft.
2. Results. The number of sequences throughout the Results should be reviewed as there are errors and the figures should be improved: RESPONSE: All the omision was edited and corrected as the peer review recomendations.
a. In line 197 “38 sequences were used as control” is incorrect, there are 34. RESPONSE: Change "34" for "38".
b. In line 204 and 205: 12 + 24 sequences are not "38". RESPONSE: Change "34" for "38".
c. Improve Figure 1 (font size and color references are missing). RESPONSE: The size of the font was increased and the figure is shared in the attached file .pdf for editing.
d. Supplementary figures must be numbered in the order in which they are cited in the text, and must have legends. Add Y-axis values ​​to charts. RESPONSE: Supplementary figures numbered in the order in which they are cited in the text, and edited adding legends Y-axis values ​​to charts.
In Results section, lines 235 to 238:
"Most of the variations were found in unique polymorphisms distributed along the mitochondrial genome (Supplementary Materials Figure S5); however, the most frequent repe-tition was concentrated in the control region, between positions 576 and 16024 of the mtDNA (Supplementary Materials Figure S4)"
It was modified according to the peer review recommendations:
"Most of the variations were found in unique polymorphisms distributed along the mito-chondrial genome (Supplementary Materials Figures S3 and S4); however, the most fre-quent repetition was concentrated in the control region, between positions 576 and 16024 of the mtDNA (Supplementary Materials Figure S5)."
e. Line 261: “From the total 86 mtDNA complete sequences”? Previously, it had been described that there were 52 complete sequences from patients with breast cancer. RESPONE: We identified 124 mtDNA sequences associated to breast cancer cases of which 86 were complete and 38 were partial sequences. Of these 86 complete sequences, 52 belonged to patients with a confirmed diagnosis of breast cancer, and 34 sequences were obtained from healthy mammary tissue of the same patients used as controls.
f. Line 269, 272 and 274: where it says Table 4, it should say Table 1. RESPONSE: Change "Table 4" for "Table 1"
g. Line 295: The data in parentheses correspond to the m.310del variant and not to the m.310T>C variant. RESPONSE: Even though the most frequent polymorphism in position 310 was m.310del, our analysis found m.310T>C in 4 of the 78 sequences (5.1%) and m.310delinsCCTC in 1 of the 78 se-quences (1.2%). After the statistical analysis using the X2 test, m.310del and m.315dup were significantly different when comparing cases and controls.
3. In the Results section a lot of discussion is added (after which there is a Discussion and Conclusions section), and in some cases they are described repeatedly.
RESPONSE: The Results section was modified and following the peer reviewer's recommendations.
4. Discussion.
a. The first 7 paragraphs of the Discussion can be shortened quite a bit.
RESPONSE: The Results section was modified and following the peer reviewer's recommendations (Paragraphs was edited and eliminate 4 lines).
b. The discussion of the difference of this work with the previous work of the authors (Ref. 60) should be expanded, since some sequences are the same.
RESPONSE: The Results section was modified and following the peer reviewer's recommendations, specifying the difference with the previous work that is still in preprint and has not been submitted
to peer review for publication, is a preliminary analysis with fewer sequences. It was edited the text on the line 454 that specifies this difference and the reference was edited correctly, adding the corresponding doi.
c. It is not highlighted what this work contributes to what is already reported on mitochondrial variants associated with breast cancer.
RESPONSE: The Results section was modified and following the peer reviewer's recommendations.
d. The fact that the 315.dup and 310del variants were found in all samples of non-European origin is not discussed or hypothesized.
RESPONSE: The Results section was modified and following the peer reviewer's recommendations. The lines 486 and 487 was added.
e. The limitations of the study should be described, for example low number of sequences analyzed, etc.
RESPONSE: The Results section was modified and following the peer reviewer's recommendations. The lines 551 and 553 was added.
5. Conclusions: they are a summary of the results. The novel contribution of the work should be emphasized.
RESPONSE: The Results section was modified and following the peer reviewer's recommendations. The lines 558 and 560 was added
All suggested changes have been made in the attached manuscript in .docx format as comments and revisions track changes that we send you for your consideration and approval, hoping to satisfy your requirements to be able to continue with the editorial process.
Greetings from Mexico
Raul C. Baptista Rosas
Universidad de Guadalajara

Reviewer 2 Report
In this paper, Hernandez et al. seek to establish associations between polymorphisms in in the control region of the mitochondrial chromosome and breast cancer. Their analysis is carried out on the information from the publicly available GenBank database. In particular, the authors point out the inconsistencies in the results previously reported in the literature, which should be attributed to erroneous experimental design, misinterpretations, and low-quality data, and seek to eliminate the respective inconsistencies.
The bioinformatics analysis accomplished by the authors is described in detail in the text. The observations made are summarized in the Results section, and further discussed. While certain dependencies were detected, several important questions related to the understanding of molecular mechanisms regulating development of breast cancer remain open, as it should have been expected.
In my opinion, this report can be helpful for the specialists in the field.
A minor issue, lines 197-199: it is stated that the control sequences were split into three groups comprising 9, 10, and 15 sequences – but this results 34 of those in total, and not 38, as indicated in the beginning of line 197.
Author Response
Dear reviewer
We appreciate all your time and effort in reviewing our manuscript and improving it for consideration by the Editorial Committee for publication.
All your comments and recommendations have been addressed as described below:
"In this paper, Hernandez et al. seek to establish associations between polymorphisms in in the control region of the mitochondrial chromosome and breast cancer.
Their analysis is carried out on the information from the publicly available GenBank database.
In particular, the authors point out the inconsistencies in the results previously reported in the literature, which should be attributed to erroneous experimental design, misinterpretations, and low-quality data, and seek to eliminate the respective inconsistencies."
RESPONSE: Reviewed and verified. We appreciate your comments on this.
"The bioinformatics analysis accomplished by the authors is described in detail in the text. The observations made are summarized in the Results section, and further discussed.
While certain dependencies were detected, several important questions related to the understanding of molecular mechanisms regulating development of breast cancer remain open, as it should have been expected.
In my opinion, this report can be helpful for the specialists in the field."
RESPONSE: Reviewed and verified. We appreciate your comments on this too.
"A minor issue mentioned in focus in lines 197-199, related to 34 control sequences 34, and not 38, as indicated in the beginning of line 197."
RESPONSE: The line 197 was edited changing "34" for "38".
All suggested changes have been made in the attached manuscript in .docx format that we send you for your consideration and approval, hoping to satisfy your requirements to be able to continue with the editorial process.
Greetings from Mexico
Raul C. Baptista Rosas
Universidad de Guadalajara

Round 2
Reviewer 1 Report
The authors have markedly improved the manuscript, fundamentally changing the objective of the work and accepting the recommendations of the reviewers. Although the results are not of a very high relevance, the present descriptive work contributes data for future deeper works.